

# Process-based and Observation-constrained SOA Simulations in China: The Role of Semivolatile and Intermediate-Volatility Organic Compounds and OH Levels

Ruqian Miao[1], Qi Chen[1,*], Manish Shrivastava[2], Youfan Chen[3], Lin Zhang[4], Jianlin Hu[5], Yan Zheng[1], Keren Liao[1]

[1]State Key Joint Laboratory of Environmental Simulation and Pollution Control, BIC-ESAT and IJRC, College of Environmental Sciences and Engineering, Peking University, Beijing, 100871, China
[2]Pacific Northwest National Laboratory, Richland, Washington, 99352, USA
[3]Sichuan Academy of Environmental policy and planning, Chengdu, Sichuan, 610041, China
[4]Laboratory for Climate and Ocean–Atmosphere Studies, Department of Atmospheric and Oceanic Sciences, School of Physics, Peking University, Beijing, 100871, China
[5]Jiangsu Key Laboratory of Atmospheric Environment Monitoring and Pollution Control, Jiangsu Engineering Technology Research Center of Environmental Cleaning Materials, Collaborative Innovation Center of Atmospheric Environment and Equipment Technology, School of Environmental Science and Engineering, Nanjing University of Information Science and Technology, Nanjing, Jiangsu, 210044, China

*Correspondence to*: Qi Chen (qichenpku@pku.edu.cn)

**Abstract.** Organic aerosol (OA) is a major component of tropospheric submicron aerosol that contributes to air pollution and causes adverse effects on human health. Chemical transport models have difficulties to reproduce the variability of OA concentrations in polluted areas, hindering understanding of the OA budget. Herein, we applied both process-based and observation-constrained schemes to simulate OA in GEOS-Chem. Comprehensive data sets of surface OA, OA components, secondary organic aerosol (SOA) precursors, and oxidants were used for model-observation comparisons. In the revised schemes, updates of the emissions, volatility distributions, and SOA yields of semivolatile and intermediate volatility organic compounds (S/IVOCs) were made. These updates are however insufficient to reproduce the SOA concentrations in observations. The addition of nitrous acid sources is an important model modification, which improves the simulation of surface concentrations of hydroxyl radical (OH) in winter in northern China. The increased surface OH concentrations enhance the SOA formation and lead to greater SOA mass concentrations by over 30%, highlighting the importance of having good OH simulations in air quality models. There is a greater sensitivity of the SOA formation to the oxidant levels in winter than in summer in China. With all the model improvements, both the process-based and observation-constrained SOA schemes can reproduce the observed mass concentrations of SOA and show spatial and seasonal consistency with each other. Our best model simulations suggest that anthropogenic S/IVOCs are the dominant source of SOA in China with a contribution of over 50%. The residential sector may be the predominant source of S/IVOCs in winter, despite large uncertainty remains in the emissions of IVOCs from the residential sector in northern China. The industry sector is also an important source of IVOCs, especially in summer. More S/IVOC measurements are needed to constrain their emissions.



## 1 Introduction

Organic aerosol (OA) is a major component of tropospheric submicron aerosol, which can be directly emitted as primary organic aerosol (POA) or formed from atmospheric oxidation processes as secondary organic aerosol (SOA) (Zhang et al., 2007). Accurate OA simulation is important for understanding the aerosol budget as well as evaluating the impacts of fine particles on air quality and human health. High OA concentrations occur in populated and polluted areas, especially in China and India (Li et al., 2017; Gani et al., 2019). However, atmospheric chemical transport models (CTMs) have difficulties in

reproducing the magnitude and the variability of OA mass in polluted environments, mainly resulting from the underestimation of SOA (Park et al., 2021; Miao et al., 2020; Jiang et al., 2019).

SOA is generally simulated in CTMs by process-based schemes, for which the oxidation of each category of lumped SOA precursors is parameterized with specific SOA yields (Chung and Seinfeld, 2002; Hodzic et al., 2016). Some of the SOA sources are uncertain. For example, the estimated annual production of anthropogenic SOA varied by tens of Tg yr$^{-1}$ in different

models, which has been attributed largely to the uncertain contribution from semivolatile and intermediate volatility organic compounds (S/IVOCs) (Spracklen et al., 2011; Hodzic et al., 2016; Pai et al., 2020). The S/IVOCs have been recognized as key SOA precursors in polluted areas for over a decade (Robinson et al., 2007; Grieshop et al., 2009). Transportation, industry, and residential use of solid fuel etc. are all important sources of S/IVOCs. Although tremendous efforts have been made to characterize their SOA production, CTMs treat their emissions, volatility distributions, reactivities, and SOA yields differently.

The emissions of S/IVOCs are estimated by applying empirical scale factors to different proxies such as POA, non-methane volatility organic compounds (NMVOCs), and speciated IVOCs (Pye and Seinfeld, 2010; Jathar et al., 2011; Shrivastava et al., 2015; Hodzic et al., 2016). The uncertainties can be over 200% for individual emission sectors, especially at a regional scale (Wu et al., 2021; Lu et al., 2020). For IVOCs, some CTMs use one lumped precursor with specific SOA yields (Pye and Seinfeld, 2010; Hodzic et al., 2016; Ots et al., 2016). Some CTMs use a volatility-basis-set (VBS) approach  for which

continuous oxidation occurs to decrease the volatility of oxidation products and alters gas-to-particle partitioning (Li et al., 2020; Chrit et al., 2018; Shrivastava et al., 2015). Although a recent study categorizes IVOCs into six groups based on volatility and molecular structure for which SOA yield parameters of each group are derived from laboratory experiments of mobile emissions (Lu et al., 2020), there is still a lack of source-dependent model frameworks.

A new observation-constrained scheme has been developed in CTMs to improve the simulation of SOA mass in polluted areas,

which estimates anthropogenic SOA formation potential based on the emission of carbon monoxide (CO) (Hodzic and Jimenez, 2011). This SOA scheme was able to reproduce the OA mass concentrations in the Mexico City metropolitan area, the United States, and China (Hodzic and Jimenez, 2011; Kim et al., 2015; Woody et al., 2016; Miao et al., 2020). The parameterization is however too generalized to differentiate specific anthropogenic source contributions. In addition, the model performance on atmospheric oxidation capacity may affect the simulation of SOA production and lead to uncertain budget analysis and source





apportionment of SOA. The measured concentrations of hydroxyl radical (OH) show high values in polluted environments in China, caused by strong production from ozone ($O_3$) and nitrous acid (HONO) as well as fast radical recycling under high concentrations of nitrogen oxide (NO) (Lu et al., 2019). For CTMs, a large model discrepancy exists in the OH simulation in the northern hemisphere (Zhao et al., 2019). Miao et al. (2020) show underestimated surface OH concentrations by over a factor of 2 at noon in winter in Beijing. The biased OH concentrations may affect the magnitude and the spatial distribution of

SOA formation in the model, which has not yet been well investigated and quantified (Feng et al., 2019; J. Zhang et al., 2019).

Herein, we conducted the OA simulations in China with both of the process-based and observation-constrained schemes in atmospheric chemical transport model GEOS-Chem. Model improvements are made on the emissions, volatility distributions, and SOA yields of S/IVOCs as well as the HONO sources. The model simulations are evaluated against nationwide measurements and the positive matrix factorization (PMF)-based source apportionment results of OA. The improved model

simulations provide insights into the control strategies of the SOA pollution in China.

## 2 Description of observations and model simulations

### 2.1 Ambient observations

The campaign-average mass concentrations of OA were taken from 68 surface measurements at urban sites, 18 measurements at suburban sites, and 8 measurements at remote sites from 2011 to 2019 (Table S1 in the Supplement). These measurements

were conducted by Aerodyne aerosol mass spectrometers (AMS) and aerosol chemical speciation monitors (ACSM) and covered mains regions in China, including North China Plain (NCP), the Yangtze River Delta (YRD), the Pearl River Delta (PRD), and Northwest China (NW). The campaign-average mass concentrations of OA factors that were resolved by PMF analysis were also synthesized. These OA factors include hydrocarbon-like OA (HOA), cooking-related OA (COA), biomass-burning-related OA (BBOA), coal-combustion-related OA (CCOA), and various oxygenated OAs (OOAs). We named the

summed concentrations of HOA, COA, BBOA, and CCOA as PMF-derived POA and those of OOAs as PMF-derived SOA. Unlike our previous study (Miao et al., 2020), we did not divide the measured concentrations by the empirical submicron-to-fine mass ratio because of the lack of such information for different regions and seasons (Y. Zheng et al., 2020; Sun et al., 2020a). Moreover, we synthesized a dataset of the campaign-average concentrations of benzene, toluene, and xylene from 49 measurements in China from 2011 to 2018 that were conducted by online gas chromatography (GC) coupled with flame

ionization detector (FID) and/or mass spectrometer (MS). Table S2 in the Supplement lists the sampling information and the results of these measurements. In addition, a recent result of primary IVOCs measured by offline sampling with thermal desorption (TD)-GC/MS in urban Shanghai (31°17′ N, 121°44′ E) from 5 December 2016 to 3 January 2017 and from 16 July to 8 August 2017 was used for comparisons in this study (Y. Li et al., 2019). We also included 28 measurements of HONO from 2011 to 2019 and 10 measurements of OH and hydroperoxy radical ($HO_2$) from 2014 to 2019 in China in the analysis

(Tables S3 and S4 in the Supplement)



## 2.2 Model configurations

Model simulations were conducted on an atmospheric chemical transport model GEOS-Chem v12.6.3 (DOI: 10.5281/zenodo.3552959) with a horizontal resolution of $0.5° \times 0.625°$ over Asia and adjacent area (11°S-55°N, 60°-150°E). The model was set for 47 vertical levels from the surface to 0.01 hPa and was driven by the MERRA2 reanalysis assimilated

meteorological data. The boundary conditions were generated by global simulations under a horizontal resolution of $2° \times 2.5°$. For computation efficiency, the model simulations for the base year of 2014 were compared with the observations. A recent study shows that the long-term trend of particulate matter is mainly driven by the change of anthropogenic emissions (Zhai et al., 2019). The emissions of nitrogen oxides ($NO_x$), NMVOCs, and organic carbon (OC) changed by $-17\%$, $+11\%$, and $-35\%$ during 2011 to 2017 in China (Zheng et al., 2018), suggesting perhaps a minor impact of the inter-annual variability on the

model evaluations herein. Common model parameters and emission inventories are described in detail elsewhere (Miao et al., 2020).

OA is simulated by so-called Complex (i.e., process-based) and Simple SOA (i.e., observation-constrained) schemes (Pai et al., 2020). The Cp_base scheme represents the default Complex SOA configuration, in which SOA is produced by the oxidation of lumped biogenic, aromatic, and S/IVOC precursors, heterogeneous uptake of glyoxal and methylglyoxal, and isoprene

multi-phase chemistry (Marais et al., 2016; Fisher et al., 2016; Pye and Seinfeld, 2010; Pye et al., 2010). The emissions of SVOCs are treated as 1.27 times of primary OC emissions, and the emissions of IVOCs are set as 66 times of naphthalene emissions (Pye and Seinfeld, 2010). Primary SVOCs are emitted as two tracers with saturation concentrations ($C^*$) of 1646 and 20 μg m$^{-3}$ (Shrivastava et al., 2006). Once emitted, SVOCs partition to the particle phase to form POA. The remaining gas-phase SVOCs are oxidized by OH with a reaction rate constant of $2 \times 10^{-11}$ cm$^3$ molec$^{-1}$ s$^{-1}$, which produces two SOA

surrogates that have two orders of magnitude lower volatilities compared to their precursors (Grieshop et al., 2009). The organic matter to OC ratios for POA and SOA are 1.4 and 2.1, respectively (Turpin and Lim, 2001). SOA produced by the oxidation of monoterpenes, sesquiterpenes, aromatics, and IVOCs is parameterized by using a VBS approach with $NO_x$-dependent SOA yields. Naphthalene is used as a surrogate of IVOCs (Chan et al., 2009). Only photooxidation is considered for aromatics and IVOCs, whereas the oxidations by OH, $O_3$, and nitrate radical ($NO_3$) are all included for monoterpenes and

sesquiterpenes (Pye et al., 2010). For isoprene, SOA is simulated by the heterogeneous uptake of isoprene oxidation products that are produced under low or high $NO_x$ conditions (Marais et al., 2016; Pai et al., 2020). The Sp_base scheme represents the default Simple SOA configuration. Primary OC emissions from the MEIC inventory are treated as non-volatile. The ratios of the emissions of anthropogenic and biomass burning surrogate precursors to CO ($EF_{SOAP}/EF_{CO}$) are fixed to 0.069 and 0.013, respectively. The SOA yields for isoprene and terpenes are set to be 0.03 and 0.10, respectively. SOA precursors are converted

to SOA with a fixed lifetime of one day (Miao et al., 2020; Pai et al., 2020).

Modifications on the SOA schemes are listed in Table 1. The Cp_R1 and Sp_R1 schemes have updates on precursor emissions, SOA yields, or parameters related to the production and removal. Specifically, the Cp_R1 scheme applies a more reasonable





scale factor of 1.0 for SVOC emissions instead of 1.27 that is used in the Cp_base scheme (Lu et al., 2018). Instead of using two bins for all sources, the volatility distributions of SVOCs emissions are specified for transportation, other anthropogenic

sources, and biomass burning and contain five bins with $C^*$ of $10^{-2}$ to $10^2$ μg m$^{-3}$ (Figure S1 in the Supplement), which have lower volatilities compared with the default distribution in the Cp_base scheme (Zhao et al., 2015; May et al., 2013b; May et al., 2013a). The updates on the emissions and SOA yields of IVOCs are described in detail in Sect. 2.3. Additionally, the scavenging efficiency of POA in wet deposition is set to be 50% instead of 0% (Shah et al., 2019). In the Sp_R1 scheme, an OH-dependent oxidation rate of SOA precursors is used for the daytime simulations, which applies a rate constant of $1.25\times10^{-11}$

135  cm$^3$ molec$^{-1}$ s$^{-1}$ instead of a fixed rate of $1.2\times10^{-5}$ s$^{-1}$ (Hodzic and Jimenez, 2011). For the nighttime simulations, a fixed oxidation rate of $2.5\times10^{-6}$ s$^{-1}$ is used instead of $1.2\times10^{-5}$ s$^{-1}$.

The Cp_R1+2 and Sp_R1+2 schemes aim at improving the OH simulation upon the Cp_R1 and Sp_R1 configurations. The GEOS-Chem model underestimates daytime surface OH concentrations in Beijing (Miao et al., 2020), which is partially driven by inadequate HONO sources. In the default model, HONO is produced by the gas-phase reaction of NO with OH as well as

the heterogeneous reaction of nitrogen dioxide (NO$_2$) on aerosols. We first revised the heterogeneous uptake coefficient of HO$_2$ ($\gamma_{HO2}$) on aerosols from 0.2 to 0.08 as suggested by Tan et al. (2020), and then added additional HONO sources in the model (Table S5 in the Supplement). Specifically, the HONO emissions from traffic sources ($E_{HONO, traffic}$) are estimated as 1.7% of the traffic NO$_x$ emissions ($E_{NOx, traffic}$) (Rappengluck et al., 2013), which can reproduce well the diurnal cycle of HONO concentrations in urban environments (Czader et al., 2015). The emissions from soil ($E_{HONO, soil}$) are estimated from the soil

NO$_x$ emissions ($E_{NOx, soil}$) by applying scale factors that depend on biomes and soil water content (Hudman et al., 2012; Oswald et al., 2013; Rasool et al., 2019). The HONO emissions from biomass burning are calculated on the basis of the burned areas provided by the Global Fire Emission Database (GFED4) and combustion-type dependent emission factors (Giglio et al., 2013; Andreae, 2019). Moreover, the heterogeneous reaction of NO$_2$ on the ground is added to the surface layer of the model. The reaction rate ($k_g$) depends on the mean molecular speed of NO$_2$ ($v_{NO2}$), the ground surface-to-volume ratio ($S_g/V$), and the

uptake coefficient of NO$_2$ on the ground ($\gamma_{g-NO2}$) (Li et al., 2010). The $S_g/V$ is set to be 0.1 m$^{-1}$ for urban areas (Vogel et al., 2003) but varies by the leaf area index and the height of the boundary layer in non-urban areas (Sarwar et al., 2008). The $\gamma_{g-NO2}$ value is set to be $10^{-6}$ for nighttime (Kurtenbach et al., 2001) and $2\times10^{-5}$ multiplied by a photo-enhancement scale factor associated with the photolysis rate of NO$_2$ ($J_{NO2}$) for daytime (J. Zheng et al., 2020). In addition, the photolysis of nitrate is considered. The photolysis rate ($J_{nitrate}$) is set to be 100 times the photolysis rate of HNO$_3$ ($J_{HNO3}$) with a HONO molar yield of

0.67 (Kasibhatla et al., 2018). Finally, in the Cp_R1+2+3 and Sp_R1+2+3 schemes, we tested the impacts of potentially underrepresented heating-season emissions of SOA precursors from the residential sector upon the previous modifications. The IVOC emissions from the residential sector during November to March are multiplied by 7 in the Cp_R1+2+3 scheme according to the observed IVOC concentrations. In the Sp_R1+2+3 scheme, the value of EF$_{SOAP}$/EF$_{CO}$ is updated from 0.069 to 0.080 for anthropogenic emissions during November to March. The factor of 0.080 has been used in other model studies for

urban plumes (Shah et al., 2019).



### 2.3 Emissions and SOA yields of IVOCs

We estimated the IVOC emissions from the emissions of NMVOCs instead of naphthalene in the revised model schemes because laboratory experiments show a better correlation of the total IVOC emissions with NMVOCs than with individual IVOC species (e.g., naphthalene) or POA (Zhao et al., 2015; Y. Zhao et al., 2016). Global anthropogenic emissions of
165 NMVOCs are provided by the Community Emissions Data System (CEDS) (Hoesly et al., 2018), and the emissions from biomass burning are provided by GFED4 (Giglio et al., 2013; Andreae, 2019). In China, anthropogenic emissions of NMVOCs are taken from the Multi-resolution Emission Inventory for China (MEIC v1.3; http://meicmodel.org). The NMVOC emission profiles of sectors (i.e., power, transportation, industry, and residential) and subsectors (i.e., gasoline, diesel, coal, solvent, biofuel or biomass burning), the IVOCs/NMVOCs emission ratios of the subsectors, and the volatility distributions of IVOCs
for the subsectors with $C^*$ of $10^6$ (IVOC6), $10^5$ (IVOC5), and $10^{\leq 4}$ (IVOC4) µg m$^{-3}$ are obtained from the literature (Figure S2 and Table S6 in the Supplement) (M. Li et al., 2019; Lu et al., 2018; Cai et al., 2019; Lim et al., 2019; Khare and Gentner, 2018). Table S7 in the Supplement lists the annual emissions of IVOC6, IVOC5, and IVOC4 in 2014. Industry and residential sectors are the major sources of IVOCs in China. The reaction rate constant with OH for these IVOC species used in the model is $2.3 \times 10^{-11}$ cm$^3$ molecule$^{-1}$ s$^{-1}$ at 298 K which is the same with the rate constant of naphthalene photooxidation (Chan et al.,
2009). Table S8 in the Supplement lists the SOA yield parametrizations of IVOCs used in this study. For high-NO$_x$ condition, mass-weighted yields of the photooxidation of C$_{12}$-C$_{14}$, C$_{15}$-C$_{16}$, and C$_{\geq 17}$ $n$-alkanes are used for IVOC6, IVOC5, and IVOC4, respectively (Presto et al., 2010; Zhao et al., 2015). For low-NO$_x$ condition, a fixed yield of 0.73 obtained from naphthalene photo-oxidation is applied to all IVOCs because of the lack of low NO$_x$ yields for $n$-alkanes (Chan et al., 2009). The corresponding IVOC yields for 10 µg m$^{-3}$ OA range from 0.19 to 0.44, which are greater than the yields in the Cp_base scheme
but within the range of the yields used in other studies (Pye and Seinfeld, 2010; Koo et al., 2014; Jathar et al., 2014; Lu et al., 2020).

Table 2 lists the total IVOC emissions estimated in various studies. Globally, the IVOC emissions range from 16.0 to 234 Tg yr$^{-1}$ for which the POA-based methods have the highest estimates and the naphthalene-based methods have the lowest (Pye and Seinfeld, 2010; Jathar et al., 2011; Shrivastava et al., 2015; Hodzic et al., 2016). Our new NMVOC-based method suggests
a global emission of 32.2 Tg yr$^{-1}$ and an emission of 6.6 Tg yr$^{-1}$ in China that is similar to the POA-based estimate made by Wu et al. (2021). The spatial distribution of IVOC emissions shows that the most increase of the new NMVOC-based emission occurs in urban areas compared with the naphthalene×66 and the POA×1.5 estimates of other models (Figure S3 in the Supplement). The POA×1.5 estimate of IVOC emissions has a greater winter-summer emission difference compared with the naphthalene×66 and the new NMVOCs-based emissions (Figure S4 in the Supplement). The additional increase of IVOC
emissions in the Cp_R1+2+3 scheme (i.e., 7 times of the residential IVOC emissions during the heating season) leads to a large emission enhancement in northern China (Figure S3) and a greater winter-summer emission difference than that in the Cp_R1+2 scheme (Figure S4), which agrees better with the PMF-derived SOA results (Sect. 3).



## 3 Results and discussion

The model performance on meteorological parameters (e.g., temperature, relative humidity, wind speed and direction, and

195 boundary layer height), oxidants (e.g., OH, $O_3$ and $NO_3$), and aerosol precursors in GEOS-Chem have been evaluated elsewhere (Miao et al., 2020). The results show the model overestimation of surface wind speed and the concentrations of $O_3$ and $NO_3$ as well as the model underestimation of boundary layer height and OH concentrations in NCP in China. Sensitivity analysis indicates that uncertainties in chemistry perhaps dominate the model biases in particulate matter and their components. The impact of the overestimated surface concentrations of $O_3$ and $NO_3$ on the SOA simulation is also minor compared with

200 the model bias of OH. We focus here on the simulations of OA. Figure 1a shows the observed campaign-average mass concentrations of OA in China which range from 0.7 to 128.5 µg m$^{-3}$. The trend of increased OA concentrations and decreased SOA mass fractions (Figure 1b) from urban to remote regions are consistent with our understanding about the primary contribution of anthropogenic sources in urban areas (Zhang et al., 2007; Li et al., 2017). The highest OA concentrations occurred in winter in northern China, corresponding to high POA fractions that may go over 50% at some urban sites. In

particular, residential solid fuel consumption emits a large amount of POA and SOA precursors and stagnant meteorological conditions often happen in winter, leading to severe haze in northern China (Li et al., 2017; Peng et al., 2019). The OA concentrations are typically low in summer when meteorological conditions favor particles dilution and deposition and in southern China where primary contributions are less than in northern China. The SOA fractions are generally high in southern China (above 65%), which may be explained by low primary emissions and high oxidation capacity that leads to fast

conversion of organic vapors to SOA (Li et al., 2015). The lowest OA concentrations were observed in remote regions (e.g., Tibetan Plateau) in summer, representing natural background conditions in China.

The statistical values such as normalized mean bias (NMB), normalized mean error (NME), root mean square error (RMSE), and Pearson's correlation coefficient ($R$) for the model-observation comparisons of campaign-average concentrations of OA, POA, and SOA are listed in Table 3. As is consistent with our previous results (Miao et al., 2020), the Cp_base simulation

underestimates the concentrations of OA (NMB = −0.46) as well as POA (NMB = −0.44) and SOA (NMB = −0.47) in China. Because a fraction of aerosol particles may present in the supermicron domain that cannot be detected efficiently by AMS or ACSM (Sun et al., 2020a), such model underestimation of OA can be greater in certain circumstances, e.g., in northern China under winter-haze conditions. By contrast, the Sp_base simulation may reproduce the OA loadings (NMB = −0.14). The POA simulations are improved in the Sp_base (NMB = −0.18) and Cp_R1 cases (NMB = −0.11). The Sp_base scheme considers

primary OC as non-volatile, for which the model results agree with the PMF-derived POA results at urban sites but significantly overestimate the POA concentrations in suburban and remotes regions (Figure S5 in the Supplement). The Cp_R1 scheme considers primary OC as semivolatile with lower volatility distributions compared with the Cp_base scheme, leading to more OC mass in the particle phase as emitted (i.e., "POA"). This scheme is slightly better than the Sp_base scheme for POA but still overestimates its concentrations in suburban and remotes regions.





Figure 1c shows the simulated SOA concentrations compared with the observations. The underestimation of the Cp_base simulation of SOA mainly occurs in urban and suburban regions, which is consistent with previous understanding about the underrepresented sources of anthropogenic SOA in process-based models (B. Zhao et al., 2016; Z. Han et al., 2016). The Sp_base simulation captures well anthropogenic SOA (NMB = −0.08) because of the use of ambient-constrained parameterization to represent anthropogenic sources (Hodzic and Jimenez, 2011; Woody et al., 2016). The $R$ increases from

0.23 in the Cp_base case to 0.65 in the Sp_base case of SOA (Table 3). For Cp_R1, the simulated SOA mass concentrations show insignificant changes (NMB = −0.49) because that the increased SOA mass from increased emissions and updated SOA yields of IVOCs is offset by the decreased SOA mass from the updated SVOC emissions and volatility distributions. The SVOC emissions and volatility distributions in Cp_R1 are supposed to be more reasonable according to recent laboratory results (Lu et al., 2018; Zhao et al., 2015; May et al., 2013b; May et al., 2013a). Further updates on HONO and residential

IVOC emissions in Cp_R1+2 and Cp_R1+2+3 improve the SOA simulations (NMB = −0.18). For observation-constrained schemes, the SOA concentrations at urban sites are lower in the Sp_R1 simulations than in the Sp_base simulations after applying OH-dependent oxidation rates for SOA precursors. This update is physically sound and may represent better the diurnal and spatial patterns of SOA formation. It is however sensitive to the simulated OH concentrations in the model. Further updates in Sp_R1+2 for HONO are therefore needed to improve the OH and subsequently the SOA simulations. The

Sp_R1+2+3 simulations demonstrate additional improvements from the potentially biased wintertime anthropogenic emissions. Figure S6 in the Supplement shows the NMB values of the SOA simulations for different seasons. The model underestimation is more significant in autumn and winter than in spring and summer. The implementation of additional HONO sources in the Cp_R1+2 and Sp_R1+2 simulations reduces the NMB values at urban and suburban sites in all seasons except in summer at urban sites. By contrast, the model modifications have a minor influence on the NMB values for remote sites in all seasons.

Figure 2a shows the ratios of seasonal mean OH concentrations simulated by the Sp_R1+2 scheme to those simulated by the Sp_R1 scheme. The modifications in Sp_R1+2 and similarly in Cp_R1+2 increase the modeled HONO concentrations (Table S3) and improve NMB from −0.58 in the base simulations to −0.14 (Figure 3). The addition of HONO sources can increase the surface mean OH concentrations by a factor of 2 to 4, especially in winter in northern China when the photolysis of HONO contributes predominantly to the primary production of OH (Tan et al., 2018; Slater et al., 2020). Table S4 lists the observed

and modeled surface OH and $HO_2$ concentrations in China. The modified HONO sources significantly improve the simulations of peak concentrations of OH and $HO_2$ in winter, which improves the SOA simulations. As shown in Figure 2b, the increased OH concentrations lead to greater SOA concentrations nationwide. In particular, the increase can be over 30% in northern China, suggesting that the SOA simulation is more sensitive to the OH simulation in northern China than in southern China. Consistently, a recent study suggests that enhanced OH levels likely promote fresh SOA formation in northern China but

increases the oxidation state of OA in southern China (J. Li et al., 2019b). In summer, the SOA mass enhancements mainly occur in the near-source regions in the Sp_R1+2 simulations. Although the OH and $HO_2$ concentrations in summer in southern and southwestern China are overestimated in the Sp_R1+2 and Cp_R1+2 simulations (Table S4), this overestimation has little





impact on the model-observation comparisons of SOA herein. Among the added HONO sources, the heterogeneous reaction of $NO_2$ on the ground contributes predominantly to the enhancements of surface HONO and OH concentrations, which is

260 consistent with the results from budget analysis of ambient observations (Xue et al., 2020; Liu et al., 2019; Huang et al., 2017). The greatest enhancements of OH concentrations therefore occur in urban areas where high $NO_x$ emissions and large $S_g/V$ facilitate the heterogeneous formation of HONO. The model parameters such as $\gamma_{g\text{-}NO2}$ and the HONO yield vary significantly by relative humidity, light intensity, and $NO_2$ concentrations and are associated with large uncertainties (C. Han et al., 2016; Liu et al., 2020), which requires more future observations to constrain.

Updates in Cp_R1+2+3 and Sp_R1+2+3 aim at increasing the emissions of anthropogenic SOA precursors. For the process-based schemes, anthropogenic aromatics, IVOCs, and SVOCs are uncertain precursors in the model. Figure 4 shows the model-observation comparison of campaign-average concentrations of benzene, toluene, and xylene in China. The model simulations in general agree with the observations with NMB of −0.31 to 0.34. The biases show insignificant seasonality, suggesting that the uncertainty in aromatic emissions is perhaps not the driven factor of the underestimated SOA concentration in winter.

Measurements of SVOCs and IVOCs are rare (Y. Li et al., 2019). Table S9 in the Supplement lists the observed and simulated campaign-average concentrations of primary IVOCs in China. The Cp_R1+2 simulation largely underestimates the IVOC concentrations, especially in winter. The underestimated IVOC emissions are likely from the residential sector that has highly uncertain emission activity (Tao et al., 2018; Peng et al., 2019; J. Li et al., 2019a). The emission factors of IVOCs from residential combustion vary in a wide range and are sensitive to the fuel types and combustion conditions (Cai et al., 2019;

Qian et al., 2021). We tested seven-fold IVOC emissions from the residential sector in the Cp_R1+2+3 simulation to eliminate the potential seasonal bias of IVOC emissions. The simulation-to-observation ratio of primary IVOC concentrations in winter became 0.44 that is similar to the ratio in summer. The Cp_R1+2+3 simulation indeed improves the winter SOA simulations at urban sites significantly and reduce NMB from −0.55 in Cp_R1+2 to −0.28 (Figure S6a).

For the observation-constrained schemes, the emissions of SOA precursors depend on the emissions of CO. The observed CO
concentrations in China are generally greater than the modeled surface concentrations, indicating possibly underestimated CO emissions especially in winter (Kong et al., 2020). Consistently, top-down estimates suggest greater CO emissions than those in the MEIC inventory (X. Zhang et al., 2019; Feng et al., 2020; Gaubert et al., 2020). On the other hand, recent measurements of SOA formation potential show a wide range of reference values for $EF_{SOAP}/EF_{CO}$ (Table S10 in the Supplement). The fixed $EF_{SOAP}/EF_{CO}$ ratios used in the model may not fully represent ambient conditions (Liao et al., 2021). In the Sp_R1+2+3

simulation, we applied a higher value of $EF_{SOAP}/EF_{CO}$ (i.e., 0.08 instead of 0.069) for all anthropogenic sources during the heating season. This modification increases the SOA concentrations in winter at urban sites and reduces NMB from −0.26 in Sp_R1+2 to −0.15 (Figure S6a). Some overestimation occurs at suburban sites (Figure S6b), highlighting the need of using source-specified $EF_{SOAP}/EF_{CO}$ in future model development. For remote sites, the increased $EF_{SOAP}/EF_{CO}$ has little influence on the model performance (Figure S6c). The spatial distributions of winter-mean SOA concentrations show greater SOA



concentrations in NCP and YRD in the Sp_R1+2+3 simulation than in the Sp_R1+2 simulation (Figure S7 in the Supplement), which agrees better with the observations.

The inter-comparisons of the process-based and observation-constrained simulations of season-mean mass concentrations of SOA are shown in Figure 5. The spatial distributions of the SOA concentrations in the Sp_R1+2 and Cp_R1+2 simulations are similar in summer with differences below 2 µg m$^{-3}$ (Figure 5a). Note that the SOA concentrations in the Sp_R1+2 and
Sp_R1+2+3 simulations are the same in summer because the updates in Sp_R1+2+3 only affect the heating season. The SOA concentrations in the Cp_R1+2 and Cp_R1+2+3 simulations in summer are also the same. In winter, the difference of the simulated SOA concentrations between Sp_R1+2+3 and Cp_R1+2 are at most 13 µg m$^{-3}$, and the main difference shows in NCP and central China where anthropogenic SOA sources are possibly still underestimated (Figure 5b). The difference of the simulated SOA concentrations between Sp_R1+2+3 and Cp_R1+2+3 is much smaller and mainly occurs in Hebei province
(Figure 5c). The spatial consistency between the Sp_R1+2+3 and Cp_R1+2+3 simulations highlights the importance of improving the estimation of IVOC emissions in China for modeling SOA. Overall, the revised process-based schemes have greater biases than the revised observation-constrained ones at urban sites (Figure S6).

Figure 6 shows the concentrations of OA, POA, and SOA as well as the mass fractions of POA and SOA simulated in the Cp_R1+2+3 simulation. The POA concentrations are several times greater in winter than in summer because of higher
emissions of S/IVOCs as well as low temperature and high OA concentrations that favor the gas-to-particle partitioning of organic vapors. The seasonal difference of SOA is smaller than that of POA. One explanation is the enhanced formation of biogenic SOA (BSOA) in summer. SOA is the dominant component of OA in summer that contributes over 60% of OA nationally, whereas POA contributes more than SOA in winter in northern China. The Cp_R1+2+3 scheme represents our best-estimate scenario for process-based simulations that capture well the seasonal and spatial patterns of OA and the split of
POA and SOA in the observations (Figure 1a,b). Figure 7 shows the corresponding OA compositions in different regions as well as their sources. The SOA mass concentrations are dominated by anthropogenic sources, among which S/IVOCs contribute over 50% in the three regions. The contribution of SVOCs to SOA depends on season. In summer, SVOC-related SOA (SVOCs-SOA) is the largest OA component in all regions, for which residential and industry sectors are the main sources. The contributions of IVOCs to OA are generally over 15% in which industry is the predominant contributor. In winter, the
residential sector is the major source of S/IVOCs. SVOCs-SOA contributes less to OA than in summer because SVOCs favorably form POA at low temperatures.

Other model studies that considered the contributions of S/IVOCs in China also show S/IVOCs contribute greatly to the simulated SOA (B. Zhao et al., 2016; Yang et al., 2019; Li et al., 2020). The mass fractions of each SOA component vary among studies. For example, Li et al. (2020) suggested a lower contribution of IVOC-related SOA (IVOCs-SOA) and a higher
contribution of aromatic SOA (ASOA) in NCP in winter compared to our study, explained by lower emissions of IVOCs and enhanced formation of ASOA from the aging process in their study. B. Zhao et al. (2016) and Yang et al. (2019) suggested over 50% contributions of IVOCs-SOA to SOA in all seasons, which is greater than our estimations. Their studies considered





the multi-generation oxidation of IVOCs for which the mechanism and parametrization remain unclear. For BSOA, its contribution to total OA is negligible in winter but can increase to 15% in PRD in summer because of the enhanced emissions

of biogenic precursors. The contribution of SOA formed by aqueous-phase ways (aqSOA) is also much greater in summer (9-13%) because high emissions of isoprene enhance the formation of IEPOX, glyoxal, and methylglyoxal (Hu et al., 2017). Field observations suggest an important role of aqSOA in SOA formation during the winter haze periods (Kuang et al., 2020; Wang et al., 2021). The simulated mass fraction of aqSOA is only 3-5% in SOA in winter herein, indicating that more precursors are perhaps involved in the SOA formation related to aerosol liquid water than the model has considered (Gkatzelis et al., 2021).

The estimated contribution of aqSOA is similar to the estimation of Li et al. (2021) in NCP but is much lower than the estimations made by Qiu et al. (2020) in Beijing and Ling et al. (2020) in PRD.

**4 Conclusions**

In this study, we applied both process-based and observation-constrained schemes to simulate OA in China. Compared with the PMF results from observations, the model underestimation of SOA mainly occurs in winter in northern China. Updated

S/IVOC emissions and SOA parameterization are insufficient to lead to the model-observation agreement. The addition of HONO sources is critical to the SOA simulations because it can significantly improve the simulations of surface OH concentrations in winter. The increased OH concentrations then enhance the SOA formation and increase the simulated SOA concentrations nationwide. Greater sensitivity of the SOA formation to the oxidant levels present in winter than in summer. In particular, the SOA concentrations may increase by over 30% in northern China in winter by improving the surface OH

simulations. With all improvements, both types of SOA schemes show seasonal and spatial variations that reasonably agree with the observations, although the revised process-based schemes have greater biases than the revised observation-constrained ones. The seasonal variations of the OA composition and the source contribution suggest that the control strategies for OA pollution should vary by season. S/IVOCs are the main contributors to OA in China with mass contributions of over 60%, highlighting the importance of controlling their emissions in haze mitigation. The model suggests the residential sector as the

major source of POA, SVOCs-SOA, and IVOCs-SOA in winter in polluted areas in China. The emissions of the residential sector however remain largely uncertain, which needs future efforts to constrain. The control of residential emissions may reduce POA and SOA simultaneously besides the reduction of other primary aerosols and secondary inorganic aerosols (Meng et al., 2020). Control measures of residential emissions have already been taken since 2017 (Lei et al., 2021; Duan et al., 2020). High OA concentrations were still observed in NCP during the COVID-19 lockdown period when the emissions from industry

and transportation were largely reduced (Sun et al., 2020b; Zheng et al., 2021), indicating the need for further reduction of residential emissions. In summer, the industry sector becomes the predominant source of S/IVOCs-SOA, which has not yet been effectively controlled in China.

*Data availability.* Data presented in this manuscript are available upon request to the corresponding author.





*Author contributions.* QC and RM designed the study. RM performed the model simulations and conducted the data analysis. LZ and YC provided the emission inventory of ammonia. QC and RM prepared the manuscript with contributions from all co-authors.

*Competing interests.* The authors declare that they have no conflict of interest.

*Acknowledgments.* This work was supported by the National Natural Science Foundation of China (41961134034, 91544107, 41875165, and 51861135102) and the 111 Project of Urban Air Pollution and Health Effects (B20009). M. Shrivastava was supported by the U.S. DOE, Office of Science, Office of Biological and Environmental Research through the Early Career Research Program.

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



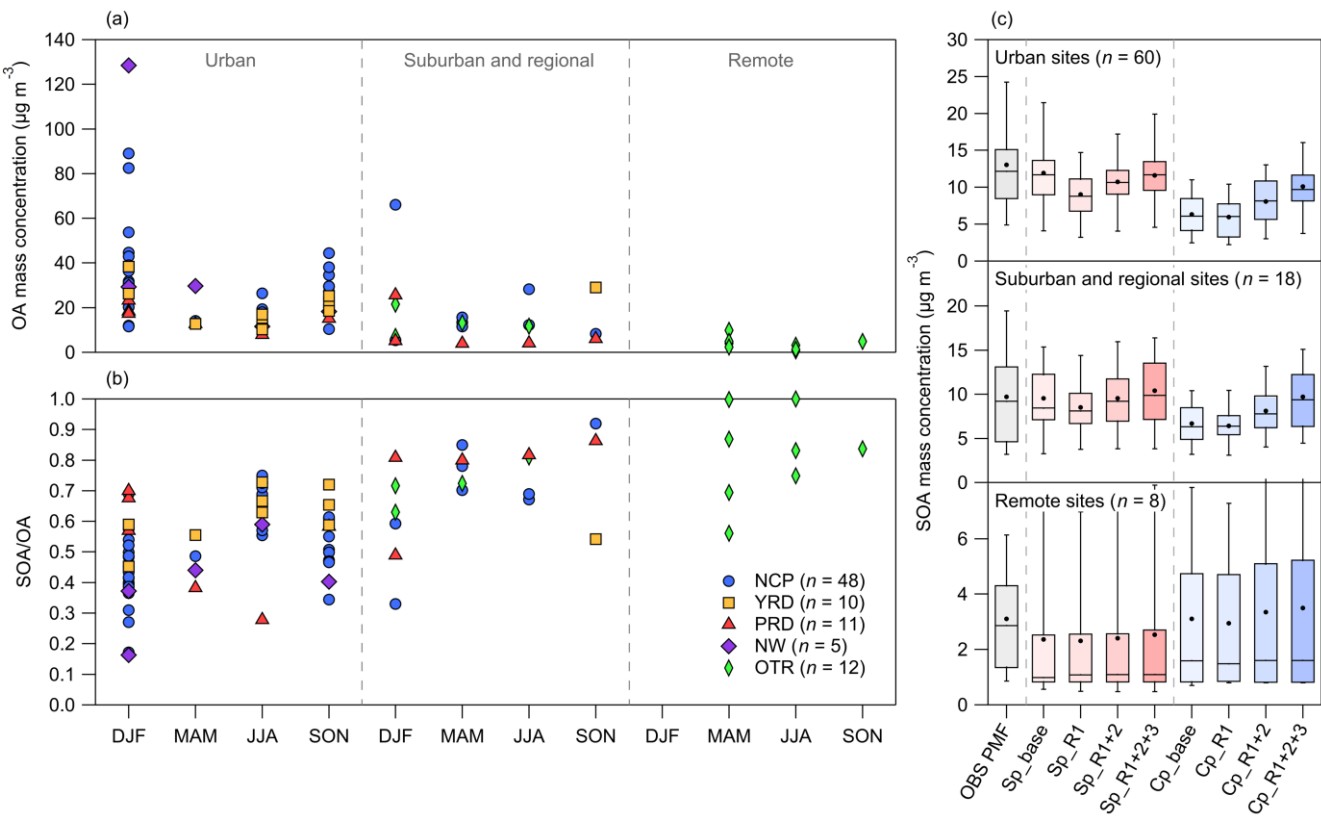

**Figure 1. (a-b)** The observed campaign-average mass concentrations of OA and the mass fraction of SOA in different seasons. **(c)** The box-and-whisker plots of the observed and simulated campaign-average mass concentrations of OA. The upper and lower edges of the boxes, the whiskers, the middle lines, and the solid dots denote the 25th and 75th percentiles, the 5th and 95th percentiles, the median values, and the mean values of the OA concentrations.

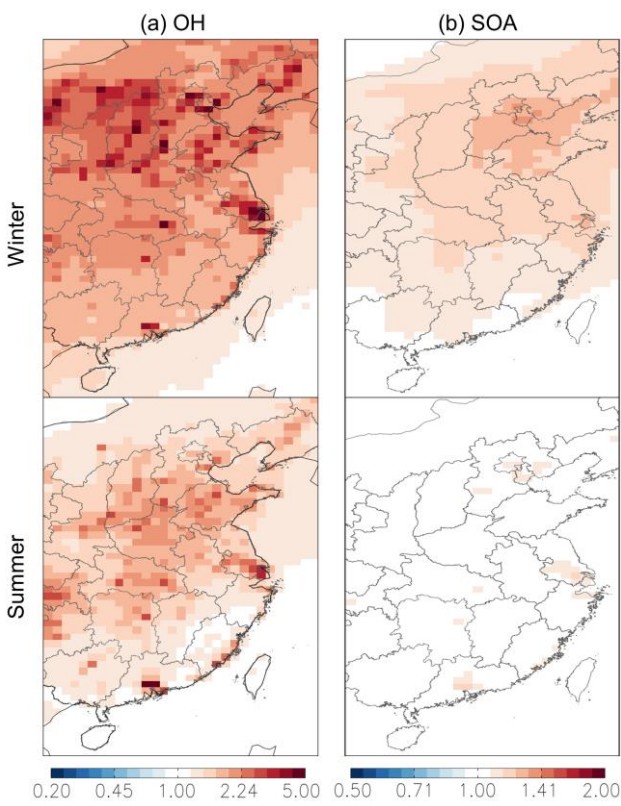

**Figure 2. The ratios of seasonal mean concentrations of (a) OH and (b) SOA simulated by the Sp_R1+2 scheme to those simulated by the Sp_R1 scheme.**



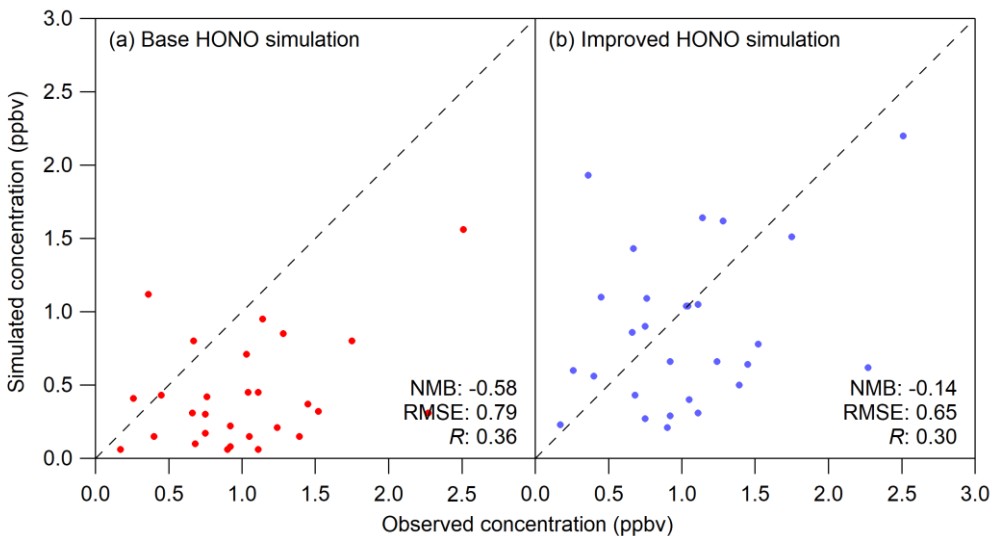

**Figure 3. Scatter plots of the observed campaign-average mixing ratios of HONO and those simulated by (a) the Sp_base scheme and (b) the Sp_R1+2 scheme. Note that the HONO mixing ratios in the Sp_base, Sp_R1, Cp_base, and Cp_R1 simulations are similar.**

**Figure 4. Comparisons of the observed campaign-average mixing ratios of benzene, toluene, and xylene in China with those simulated by the Cp_base scheme. Note that the mixing ratios of these aromatic compounds in the Cp_base, Cp_R1, Cp_R1+2, and Cp_R1+2+3 simulations are similar.**

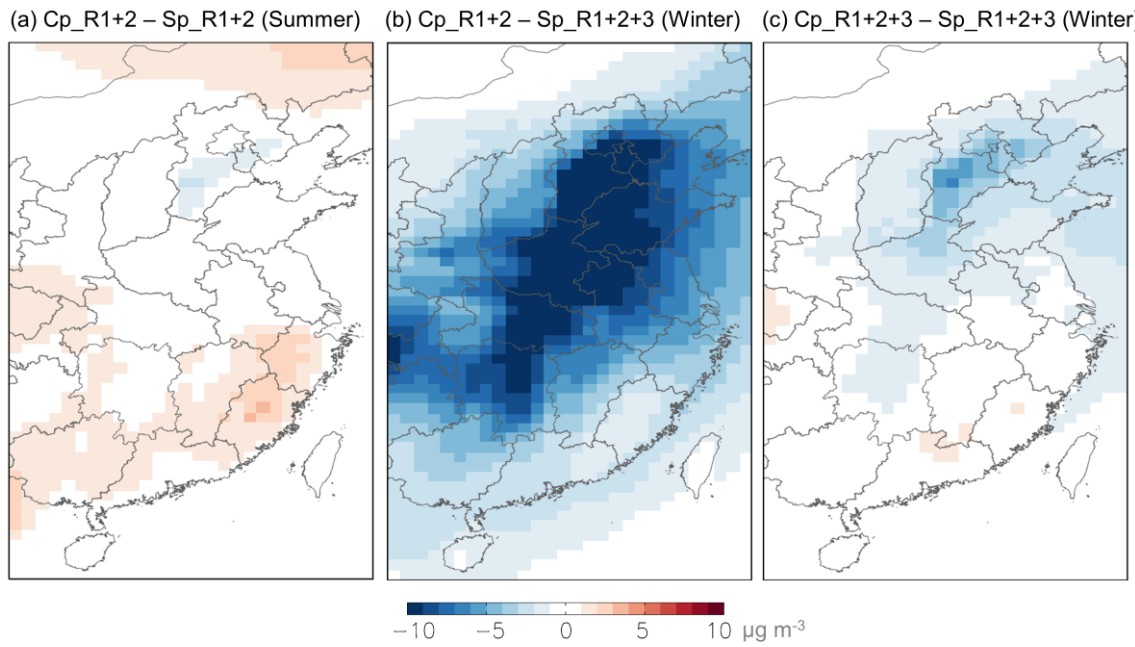

**Figure 5. The differences of seasonal mean mass concentrations of SOA between (a) the Cp_R1+2 and Sp_R1+2 simulations in summer, (b) the Cp_R1+2 and Sp_R1+2+3 simulations in winter, and (c) the Cp_R1+2+3 and Sp_R1+2+3 simulations in winter.**

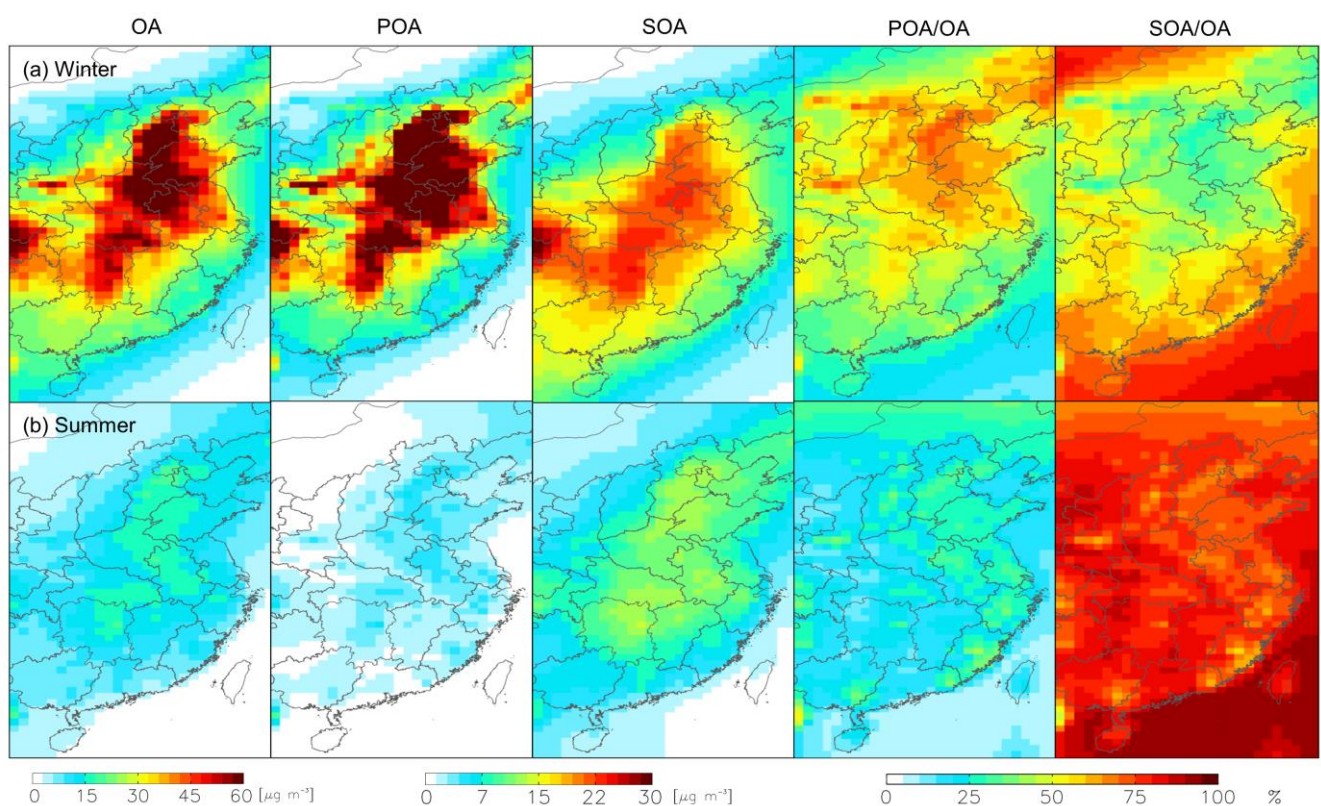

**Figure 6. Seasonal mean mass concentrations of OA, POA, and SOA and the mass fractions of POA and SOA simulated by the Cp_R1+2+3 scheme in (a) winter and (b) summer.**





**Figure 7. Seasonal mean mass fractions of OA components and the sources of POA, SVOCs-SOA, and IVOCs-SOA in NCP, YRD, and PRD in (a) winter and (b) summer simulated by the Cp_R1+2+3 scheme.**



**Table 1. Descriptions of the OA simulations in this study. Additional HONO sources include direct emissions from traffic, soil, and biomass burning as well as secondary formation from the heterogeneous reaction of NO₂ on the ground and the photolysis of nitrate.**

| | Modifications | Cp_base | Cp_R1 | Cp_R1+2 | Cp_R1+2+3 |
|---|---|---|---|---|---|
| Complex SOA (process-based) scheme | Updated emissions of S/IVOCs and SOA yields of IVOCs | | ○ | ○ | ○ |
| | Additional HONO sources and lower $\gamma_{HO2}$ | | | ○ | ○ |
| | Increased IVOC emissions from the residential sector during the heating season | | | | ○ |
| | Modifications | Sp_base | Sp_R1 | Sp_R1+2 | Sp_R1+2+3 |
| Simple SOA (observation-constrained) scheme | OH-dependent oxidation rate of SOA precursors | | ○ | ○ | ○ |
| | Additional HONO sources and lower $\gamma_{HO2}$ | | | ○ | ○ |
| | Increased $EF_{SOAP}/EF_{CO}$ during the heating season | | | | ○ |



**Table 2. The annual emissions of IVOCs derived by different approaches in literature and in our study. The values in parentheses are the IVOCs emissions when the residential emission is multiplied by a factor of 7.**

| Region | Literature | Base year | Methods | Emission inventories of surrogates[*] | | IVOC emissions (Tg yr⁻¹) |
| | | | | Anthropogenic | Biomass burning | |
|---|---|---|---|---|---|---|
| World | Pye and Seinfeld (2010) | 2000 | Naphthalene×66 | EDGAR2 Zhang and Tao (2009) | GFED2 | 16.0 |
| | Jathar et al. (2011) | 2000 | POA×1.5 | Bond et al. (2004) | GFED2 | 84.6 |
| | Shrivastava et al. (2015) | 2000 | POA×6.5 | IPCC-AR5 | GFED3 | 234 |
| | Hodzic et al. (2016) | 2000 | NMVOCs×0.2 | RETRO | GFED3 | 19.7 |
| | This study | 2014 | Naphthalene×66 | Shen et al. (2013) | Shen et al. (2013) | 16.2 |
| | | 2014 | NMVOCs-based Sector and subsector specified | CEDS | GFED4 | 32.2 |
| China | B. Zhao et al. (2016) | 2010 | Gasoline: POA×30.0 Diesel: POA×4.5 Biomass burning: POA×1.5 Other sources: POA×3.0 | Wang et al. (2014) | Not included | 10.1 |
| | Wu et al. (2021) | 2016 | Industry, transportation, and power: POA×scale factor (mean: 8.39) Residential, shipping, and biomass burning: POA×scale factor (mean: 0.43) | MEIC | FINN | 6.7 |
| | This study | 2014 | Naphthalene×66 | Shen et al. (2013) | Shen et al. (2013) | 3.8 |
| | | | POA×1.5 | MEIC | GFED4 | 5.7 |
| | | | NMVOCs-based Sector and subsector specified | MEIC | GFED4 | 6.6 (11.0) |
| NCP China | Li et al. (2020) | 2015 | Transportation: Liu et al. (2017) Industry and residential: POA×0.34(low)/1.5(medium) | MEIC Liu et al. (2017) | Not included | 0.1-0.4 |
| | This study | 2014 | NMVOCs-based Sector and subsector specified | MEIC | GFED4 | 0.7 (1.0) |
| YRD China | Huang et al. (2021) | 2017 | Transportation: POA×8.0 Other sources: POA×1.5 | MEIC | Not included | 0.7 |
| | | | EF-based | MEIC | Not included | 0.3 |
| | This study | 2014 | NMVOCs-based Sector and subsector specified | MEIC | GFED4 | 0.9 (1.2) |

[*]: Emissions Database for Global Atmospheric Research version 2 (EDGAR2); Intergovernmental Panel on Climate Change Fifth Assessment Report emission data set (IPCC-AR5); REanalysis of the TROpospheric chemical composition emission inventory (RETRO)





**Table 3. The statistics of model-observation comparisons of the campaign-average mass concentrations of OA, POA, and SOA in China. "OBS" and "SIM" represent the mean values of the observations and the simulations, respectively. The units of OBS, SIM, and RMSE are µg m$^{-3}$.**

|     |      | Cp_base | Cp_R1 | Cp_R1+2 | Cp_R1+2+3 | Sp_base | Sp_R1 | Sp_R1+2 | Sp_R1+2+3 |
|-----|------|---------|-------|---------|-----------|---------|-------|---------|-----------|
| OA  | OBS  |         |       |         | 22.99     |         |       |         |           |
|     | SIM  | 12.42   | 15.86 | 17.28   | 19.18     | 19.84   | 17.60 | 19.00   | 19.82     |
|     | NMB  | -0.46   | -0.31 | -0.25   | -0.17     | -0.14   | -0.23 | -0.17   | -0.14     |
|     | NME  | 0.52    | 0.44  | 0.41    | 0.39      | 0.36    | 0.42  | 0.40    | 0.39      |
|     | RMSE | 18.03   | 15.76 | 15.00   | 14.31     | 13.91   | 15.52 | 14.74   | 14.47     |
|     | *R*  | 0.71    | 0.72  | 0.72    | 0.71      | 0.73    | 0.69  | 0.70    | 0.70      |
| POA | OBS  |         |       |         | 11.51     |         |       |         |           |
|     | SIM  | 6.47    | 10.19 | 9.77    | 9.90      | 9.42    | 9.41  | 9.41    | 9.41      |
|     | NMB  | -0.44   | -0.11 | -0.15   | -0.14     | -0.18   | -0.18 | -0.18   | -0.18     |
|     | NME  | 0.58    | 0.45  | 0.45    | 0.45      | 0.50    | 0.50  | 0.50    | 0.50      |
|     | RMSE | 11.31   | 9.86  | 9.88    | 9.88      | 10.60   | 10.60 | 10.60   | 10.60     |
|     | *R*  | 0.74    | 0.74  | 0.75    | 0.75      | 0.73    | 0.73  | 0.73    | 0.73      |
| SOA | OBS  |         |       |         | 11.41     |         |       |         |           |
|     | SIM  | 6.10    | 5.78  | 7.62    | 9.39      | 10.53   | 8.30  | 9.70    | 10.51     |
|     | NMB  | -0.47   | -0.49 | -0.33   | -0.18     | -0.08   | -0.27 | -0.15   | -0.08     |
|     | NME  | 0.52    | 0.55  | 0.45    | 0.39      | 0.34    | 0.40  | 0.37    | 0.37      |
|     | RMSE | 8.36    | 8.77  | 7.34    | 6.08      | 5.27    | 6.54  | 5.67    | 5.53      |
|     | *R*  | 0.23    | 0.13  | 0.32    | 0.50      | 0.65    | 0.48  | 0.57    | 0.59      |