# Peer review of "Process-based and Observation-constrained SOA Simulations in China: The Role of Semivolatile and Intermediate-Volatility Organic Compounds and OH Levels"

_Atmospheric Chemistry and Physics, 2021_

## Author Comment (AC1)

**Response to reviews**

Reviewer comments are in **bold**. Author responses are in plain text labeled with [R]. Line numbers in the responses correspond to those in the revised manuscript (the track-change version). Modifications to the manuscript are in *italics*.

**Reviewer #1**

**Miao and coauthors applied process-based and observation-constrained schemes to simulate OA in China. They found that the addition of nitrous acid sources and therefore the enhanced OH concentration are critical for the improved performance of model simulation. Overall, it is scientifically valid work and advances the important work of simulating SOA in polluted areas. My primary concern is that the bulk of science here is major in the "Model techniques and comparison" not the "Chemistry and Physics". With its current version, I am not totally convinced that it should be published in this specific journal. Also, I do have some technical concerns noted in my general comments below.**

[R0] We thank the reviewer for the valuable feedback. The paper provides important information for a better understanding the budget and sources of OA in China as well the sensitivity of SOA formation to OH levels. We have clarified the scientific points of the paper and downweighted the technical sound by various changes in the Abstract, Introduction, Section 3, and Conclusions. Detailed responses to the comments are given below.

**(1) To evaluate the model performance, the authors compared the OA simulations with campaign-average OA observations from 2011 to 2019. Do the authors run model simulations also from 2011 to 2019? It was claimed that the base year of the model simulation was 2014 (Line 101). Then for comparison, how do the authors match their simulations with observations from different periods and different years? It seems that the authors ignored the impact of inter-annual variability in their study. But both the emission inventory and observations has demonstrated significant changes of organic carbon in China due to clean air actions. How do the authors consider this concern?**

[R1] The model simulations were run for the year of 2014 and sampled the time and location of each campaign except for the specific year. We think the inter-annual variability would not affect the analysis and conclusions in the manuscript. Recent studies have shown that the inter-annual variability of particulate matter in China is mainly driven by the change of anthropogenic emissions (Geng et al., 2021; Zhai et al., 2019). The change of anthropogenic NMVOCs emissions (e.g., SOA precursors) in China is rather small over years (+11% from 2011 to 2017). We agree with the reviewer that the organic carbon (OC) emission in China has been significantly reduced since 2014 due to clean air actions. For the OA simulations, primary OC affects the SOA concentrations via loading-dependent gas-to-particle partitioning, which is expected to be small (roughly < 10%) given the range of POA loadings (e.g., about 10-35 µg m$^{-3}$ in NCP (Duan et al., 2020)) and the volatility distributions of semi-volatile organic vapors (Xu et al., 2019; Xu et al., 2021). Also, the majority of the observation data used in this study are from 2013 to 2015 (52/86) during which the changes of emissions that are related to OA simulations are even smaller.

To clarify the potential effects of inter-annual variability, we have revised Line 123-134 as follows: "*For computation efficiency, the model simulations were run for the year of 2014 and sampled the time and*

*location of each campaign except for the specific year for model-observation comparisons. Recent studies show that the long-term trend of particulate matter is mainly driven by the change of anthropogenic emissions in China (Zhai et al., 2019; Geng et al., 2021). The emissions of nitrogen oxides ($NO_x$), NMVOCs, and organic carbon (OC) changed by −17%, +11%, and −35% from 2011 to 2017 (Zheng et al., 2018), suggesting a minor emission change of anthropogenic SOA precursors over years. The change of primary OC emission is significant and can reduce the surface POA concentrations (e.g., 20-30 µg m$^{-3}$ as observed in NCP) (Duan et al., 2020). Its impact on SOA concentrations due to loading-dependent gas-particle partitioning is however less than 10% given the OA mass loadings and the volatility distributions of semivolatile organic vapors (e.g., with mean saturation concentrations of 0.5-0.75 µg m$^{-3}$ in Beijing) (Xu et al., 2019; Xu et al., 2021). The majority of the surface OA observations (52 out of 86 in Table S1) is from 2013 to 2015 during which the emission changes related to OA are even smaller*".

**(2) The authors ignored the difference between submicron and fine particles due to the lack of information (Line 86). How would it influence the model-observation comparison and thus conclusions from this study?**

[R2] The model simulations herein all underestimate the SOA concentrations. Taking into account the potential missing mass in the supermicron domain, the underestimation would become greater as stated in Line 252-254. It would not affect the conclusions in the paper about the significant impact of OH on SOA and the main contribution of S/IVOCs to SOA. For clarification, we have revised Line 101-102 as follows: "*Because the model simulations consistently underestimate the OA concentrations, taking into account the potential supermicron mass may lead to greater model-observation gaps but not affect the analysis herein*".

**(3) The SOA yields for monoterpenes and sesquiterpenes were set to be the same in this study (Line 124). But lots of studies have shown significantly different SOA yields of MT and SQT (Yee et al., 2018, Atmos. Chem. Phys., and references therein). Would this largely vary the SOA simulations, especially in summer?**

[R3] The Complex SOA schemes have considered different SOA yields for MT and SQT. In Simple SOA schemes, the SOA simulation is greatly simplified and therefore a single yield of 0.1 is used for both MT and SQT. This yield is more close to the expected yield of MT for the OA loadings in China and thus may underestimate the contribution of SQT to SOA in Simple SOA schemes. However, the contribution of SQT SOA in China is small compared with other types of biogenic SOA and negligible compared with anthropogenic SOA, even in summer in South China where biogenic VOC emissions are high (Figure 7). Therefore, this yield would not affect much the SOA simulations in our study. Even in biogenic-dominant areas in US and the Simple SOA schemes reproduce the observation well (Pai et al., 2020). We have revised Line 153-156 to clarify this part as follows: "*The SOA yields for isoprene and terpenes are set to be 0.03 and 0.10, respectively, for simplification. Good model performance has been found for biogenic-dominant regions in the U.S., indicating such simplified yield parameterization works in ambient environments, although the yields for terpenes observed in the laboratory can be quite different (Pai et al., 2020)*".

**(4) In addition to the statistical values, please also provide the scatter plots for model-observation comparisons with 1:1 line. To clearly see how the model performance varied for urban, suburban, and remote regions, the authors can set different colors for the data points.**

[R4] We have added Figure S6 in SI to show the scatter plots for the model evaluation of SOA and described it in Line 261.

**(5) In "Results and discussion", the authors mainly focused on the comparison of different model schemes but not the scientific information on chemistry and physics. I suggest to separate this part into several sections and focus more on the scientific value of their model results.**

[R5] We thank the reviewer for the suggestion. We have added subsections in Sect. 3 and revised the text to clarify the scientific points of the paper as replied in [R0].

**Other comments and suggestions to the text:**

**Line 105: While more details can be found elsewhere, please still briefly describe in this study.**

[R6] We have added the descriptions of model setting in Line 116-123 as follows: "*The model simulated the ozone–$NO_x$–hydrocarbon–aerosol chemistry with ISORROPIA-II thermodynamic equilibrium model (Park et al., 2004; Fountoukis and Nenes, 2007). Global emissions for anthropogenic, biogenic, and biomass burning were provided by the Community Emissions Data System (CEDS) (Hoesly et al., 2018), the Model of Emissions of Gases and Aerosols from Nature (MEGAN v2.1) (Guenther et al., 2012), and the emissions from biomass burning are provided by the Global Fire Emission Database (GFED4) (Giglio et al., 2013), respectively. In China, anthropogenic emissions were taken from Zhang et al. (2018) for ammonia and the Multi-resolution Emission Inventory for China (MEIC v1.3; http://meicmodel.org) for other pollutants. More details of the model settings are provided our previous study (Miao et al., 2020)*".

**Line 125: Explain more here why it makes sense to use a fixed lifetime for all SOA precursors.**

[R7] We have revised Line 156-158 as follows: "*SOA precursors are converted to SOA with a fixed lifetime of one day (Miao et al., 2020; Pai et al., 2020), which generally represent the e-folding timescale of the SOA formation observed in polluted environments (DeCarlo et al., 2010; Hayes et al., 2013)*".

**Line 136: Why the authors use $2.5\times10^{-6}$ s$^{-1}$? Reasons or references?**

[R8] The OH measurements in China showed that the campaign-averaged nighttime OH concentrations are in the range of $0.2$-$0.8\times10^6$ molecules cm$^{-3}$ (Slater et al., 2020; Whalley et al., 2021; Yang et al., 2021, and references therein). This is much smaller than the daytime OH concentrations, suggesting that the fixed oxidation rate of $1.2\times10^{-5}$ s$^{-1}$ is too high for nighttime SOA formation. It is also inappropriate to use the OH-dependent oxidation rate as the same as the daytime parameterization for nighttime SOA formation because the nighttime oxidation to lead to SOA formation is related to $NO_3$ and $O_3$. We therefore use the fixed oxidation rate of $2.5\times10^{-6}$ s$^{-1}$o roughly represent the $NO_3$ and $O_3$ oxidation of SOA precursors, which is equivalent to the oxidation rate under an OH level of $0.2\times10^6$ molecules cm$^{-3}$. We have revised Line 168-171 as follows: "*For the nighttime simulations, a fixed oxidation rate of $2.5\times10^{-6}$ s$^{-1}$ is used instead of $1.2\times10^{-5}$ s$^{-1}$ to account for the $NO_3$ and $O_3$ oxidation at night, which is equivalent to the daytime oxidation rate for an OH level of $0.2\times10^6$ molecules cm$^{-3}$ (Slater et al., 2020; Whalley et al., 2021; Yang et al., 2021, and references therein)*".

**Line 196: How much does the model overestimate $O_3$ and $NO_3$ concentrations and underestimate OH concentration? Please provide the specific number (e.g., %) and their potential impacts on SOA simulations.**

[R9] The model underestimates the peak concentrations of OH by a factor of 2–4 in northern China (Line

288). By contrast, the model overestimates the peak concentrations of $O_3$ in winter and $NO_3$ in summer by 2 and 3 times, respectively (Miao et al., 2020). The biases in the simulations of $O_3$ and $NO_3$ mainly affect the formation of biogenic SOA (Pye et al., 2010) while the bias in the OH simulation affect the SOA formation from all precursors. We have revised Line 286-293 as follows: "*The results show the model overestimation of surface wind speed, the peak $O_3$ concentrations by a factor of 2 in winter, and the peak $NO_3$ concentrations by a factor of 3 in summer. On the other hand, the model underestimates the boundary layer height and the daytime surface OH concentrations by a factor of 2-4 in NCP in China. Sensitivity analysis indicates that uncertainties in chemistry dominate the model biases in particulate matter and its components. The impact of the overestimated surface concentrations of $O_3$ and $NO_3$ on the SOA simulation is probably minor compared with the model bias of OH because OH is the dominant oxidant in China and the influences of $O_3$ and $NO_3$ limit to the formation of biogenic SOA (BSOA) that is a minor contributor to the SOA mass compared with anthropogenic sources in polluted environments (Zhu et al., 2020; Pye et al., 2010)*".

**References**

Duan, J., Huang, R.-J., Li, Y. J., Chen, Q., Zheng, Y., Chen, Y., Lin, C., Ni, H., Wang, M., Ovadnevaite, J., Ceburnis, D., Chen, C., Worsnop, D. R., Hoffmann, T., O'Dowd, C., and Cao, J.: Summertime and wintertime atmospheric processes of secondary aerosol in Beijing, Atmos. Chem. Phys., 20, 3793-3807, https://doi.org/10.5194/acp-20-3793-2020, 2020.

Geng, G., Zheng, Y., Zhang, Q., Xue, T., Zhao, H., Tong, D., Zheng, B., Li, M., Liu, F., Hong, C., He, K., and Davis, S. J.: Drivers of $PM_{2.5}$ air pollution deaths in China 2002-2017, Nat. Geosci., 14, 645-650, https://doi.org/10.1038/s41561-021-00792-3, 2021.

Pai, S. J., Heald, C. L., Pierce, J. R., Farina, S. C., Marais, E. A., Jimenez, J. L., Campuzano-Jost, P., Nault, B. A., Middlebrook, A. M., Coe, H., Shilling, J. E., Bahreini, R., Dingle, J. H., and Vu, K.: An evaluation of global organic aerosol schemes using airborne observations, Atmos. Chem. Phys., 20, 2637-2665, https://doi.org/10.5194/acp-20-2637-2020, 2020.

Pye, H. O. T., Chan, A. W. H., Barkley, M. P., and Seinfeld, J. H.: Global modeling of organic aerosol: the importance of reactive nitrogen ($NO_x$ and $NO_3$), Atmos. Chem. Phys., 10, 11261-11276, https://doi.org/10.5194/acp-10-11261-2010, 2010.

Slater, E. J., Whalley, L. K., Woodward-Massey, R., Ye, C., Lee, J. D., Squires, F., Hopkins, J. R., Dunmore, R. E., Shaw, M., Hamilton, J. F., Lewis, A. C., Crilley, L. R., Kramer, L., Bloss, W., Vu, T., Sun, Y., Xu, W., Yue, S., Ren, L., Acton, W. J. F., Hewitt, C. N., Wang, X., Fu, P., and Heard, D. E.: Elevated levels of OH observed in haze events during wintertime in central Beijing, Atmos. Chem. Phys., 20, 14847-14871, https://doi.org/10.5194/acp-20-14847-2020, 2020.

Whalley, L. K., Slater, E. J., Woodward-Massey, R., Ye, C., Lee, J. D., Squires, F., Hopkins, J. R., Dunmore, R. E., Shaw, M., Hamilton, J. F., Lewis, A. C., Mehra, A., Worrall, S. D., Bacak, A., Bannan, T. J., Coe, H., Percival, C. J., Ouyang, B., Jones, R. L., Crilley, L. R., Kramer, L. J., Bloss, W. J., Vu, T., Kotthaus, S., Grimmond, S., Sun, Y., Xu, W., Yue, S., Ren, L., Acton, W. J. F., Hewitt, C. N., Wang, X., Fu, P., and Heard, D. E.: Evaluating the sensitivity of radical chemistry and ozone formation to ambient VOCs and $NO_x$ in Beijing, Atmos. Chem. Phys., 21, 2125-2147, https://doi.org/10.5194/acp-21-2125-2021, 2021.

Xu, W., Xie, C., Karnezi, E., Zhang, Q., Wang, J., Pandis, S. N., Ge, X., Zhang, J., An, J., Wang, Q., Zhao, J., Du, W., Qiu, Y., Zhou, W., He, Y., Li, Y., Li, J., Fu, P., Wang, Z., Worsnop, D. R., and Sun, Y.: Summertime aerosol volatility measurements in Beijing, China, Atmos. Chem. Phys., 19, 12, https://doi.org/10.5194/acp-19-10205-2019, 2019.

Xu, W., Chen, C., Qiu, Y., Li, Y., Zhang, Z., Karnezi, E., Pandis, S. N., Xie, C., Li, Z., Sun, J., Ma, N., Xu, W., Fu, P., Wang, Z., Zhu, J., Worsnop, D. R., Ng, N. L., and Sun, Y.: Organic aerosol volatility and viscosity in the North China Plain: contrast between summer and winter, Atmos. Chem. Phys., 21, 5463-5476, https://doi.org/10.5194/acp-21-5463-2021, 2021.

Yang, X., Lu, K., Ma, X., Liu, Y., Wang, H., Hu, R., Li, X., Lou, S., Chen, S., Dong, H., Wang, F., Wang, Y., Zhang, G., Li, S., Yang, S., Yang, Y., Kuang, C., Tan, Z., Chen, X., Qiu, P., Zeng, L., Xie, P., and Zhang, Y.: Environmental Research Letters, Sci. Total Environ., 772, 144829, https://doi.org/10.1016/j.scitotenv.2020.144829, 2021.

Zhai, S., Jacob, D. J., Wang, X., Shen, L., Li, K., Zhang, Y., Gui, K., Zhao, T., and Liao, H.: Fine particulate matter ($PM_{2.5}$) trends in China, 2013–2018: separating contributions from anthropogenic emissions and meteorology, Atmos. Chem. Phys., 19, 11031-11041, https://doi.org/10.5194/acp-19-11031-2019, 2019.

---

## Author Comment (AC2)

**Response to reviews**

Reviewer comments are in **bold**. Author responses are in plain text labeled with [R]. Line numbers in the responses correspond to those in the revised manuscript (the version with all changes accepted). Modifications to the manuscript are in *italics*.

**Reviewer #2**

**The work improves SOA simulations by both of process-based and observation-constrained schemes. The authors clarify all updates in revised model simulations and highlight an important model modification, namely the addition of nitrous acid sources. The model shows a good correlation with the observations in different regions and seasons, giving confidence that there is value in the technique. The paper not only presents a reasonable way of improving SOA simulations, but also uses it to interpret air quality sources and phenomena in China. The authors then go on to make source analysis and provide insights into haze mitigation. The paper is good that it offers further evidence that the importance of controlling residential emissions in winter in polluted areas in China. Overall, the quality of English is good. As such, I think this MS can be accepted.**

[R0] We thank the reviewer for the valuable feedback. To further improve the paper, we have clarified the scientific points including the budget and sources of OA in China as well the sensitivity of SOA formation to OH levels and downweighted the technical sound by various changes in the Abstract, Introduction, Section 3, and Conclusions in the revised version.